# PI3k inhibitors (BKM120 and BYL719) as radiosensitizers for head and neck squamous cell carcinoma during radiotherapy

Fu-Cheng Chuang[1,2], Chih-Chun Wang[2,3], Jian-Han Chen[2,4], Tzer-Zen Hwang[2,3], Shyh-An Yeh[1,5], Yu-Chieh Su[2,6]*

1 Department of Radiation Oncology, E-Da Hospital, Kaohsiung, Taiwan, 2 Department of Medicine, School of Medicine, I-Shou University, Kaohsiung, Taiwan, 3 Department of Otolaryngology, E-Da Hospital, Kaohsiung, Taiwan, 4 Department of General Surgery, E-Da Hospital, Kaohsiung, Taiwan, 5 Department of Medical Imaging and Radiological Sciences, I-Shou University, Kaohsiung City, Taiwan, 6 Division of Hematology-Oncology, Department of Internal Medicine, E-Da Hospital, Kaohsiung, Taiwan

* hepatoma@gmail.com

**Data Availability Statement:** All relevant data are within the manuscript.

**Funding:** The authors received no specific funding for this work.

## Abstract

Approximately 500,000 new cases of head and neck squamous cell carcinoma (HNSCC) are reported annually. Radiation therapy is an important treatment for oral squamous cell carcinoma (OSCC). The survival rate of patients with HNSCC remained low (50%) in decades because of radiation therapy failure caused by the radioresistance of HNSCC cells. This study aimed to identify PI3K inhibitors that can enhance radiosensitivity. Results showed that pan-Phosphoinositide 3-kinases (PI3K) inhibitor BKM120 and class I α-specific PI3K inhibitor BYL719 dose-dependently reduced the growth of OSCC cells but not that of radioresistant OML1-R cells. The combination treatment of BKM120 or BYL719 with radiation showed an enhanced inhibitory effect on OSCC cells and radioresistant OML1-R cells. Furthermore, the enhanced inhibitory effect of the combination treatment was confirmed in patient-derived OSCC cells. The triple combination treatment of mTOR inhibitor AZD2014 and BKM120 or AZD2014 and BYL719 with radiation showed a significantly enhanced inhibitory effect on radioresistant OML1-R cells. These results suggest that the PI3K inhibitors are potential therapeutic agents with radiosensitivity for patients with OSCC.

## Introduction

Head and neck squamous cell carcinoma (HNSCC) is the sixth most common type of cancer worldwide, remaining a public health concern with an estimated annual incidence of 500,000 new cases [1]. Approximately 90% head and neck cancer are squamous cell carcinoma and appear in upper aerodigestive tract organs, such as oral cavity, larynx, and pharynx [2]. Despite the improvement in HNSCC diagnosis and treatment through radiation, surgery, chemotherapy, concurrent chemoradiation, and monoclonal antibodies, the 5-year survival of patients with HNSS remains low at 40% [3]. Locoregional, distant relapse, and regional recurrences resistant to tumor therapies are the main reason of death in patients with HNSCC [4]. Recurrent tumors arise from radiation-resistant carcinoma cells, leading to re-irradiation treatment failure [5]. However, the combined administration of chemotherapeutics and radiation

**Competing interests:** The authors have declared that no competing interests exist.

therapy is accompanied with severe toxic side effects that lead to the limited overall survival of patients [6]. Therefore, effective agents or improving the sensitization of radiation strategies is important for the treatment of these patients.

The phosphatidylinositol 3-kinase (PI3K) signaling pathway regulates the proliferation, survival, and apoptosis of cells [7]. The genetic aberration or dysregulation of the components involved in the PI3K signaling pathway, such as AKT and PTEN, is associated with the development and progression of cancers including HNSCC; it influences the metabolism, tumor growth, and development of metastasis [8]. Furthermore, the PI3K pathway is associated with resistance to anticancer therapies including targeted drugs, radiotherapy, and chemotherapy [9]. However, inhibitors targeting the PI3K/AKT/mTOR signaling pathway reportedly restore sensitivity to inhibit tumor growth under a combination treatment [10]. PI3K inhibitors also increase sensitivity in tumors resistant to chemotherapy, radiation, and hormone therapy [11].

Alpelisib (BYL719) is a class I α-specific PI3K inhibitor with strong inhibitory activity to wild-type and mutant PI3Kα isoforms but with weak inhibition to PI3K β, γ, and δ isoforms [12]. The combination treatment of BYL719 and epidermal growth factor receptor inhibitor synergistically inhibits the growth of HNSCC cells [13]. Buparlisib (BKM120) is a potent pan-PI3K inhibitor that inhibits all class IA PI3K paralogs [14]. Preclinical evidence in HNSCC has shown that BYL719 and BKM120 are under clinical trials. The present study analyzed the radiosensitivity of BYL719 and BKM120 in oral squamous carcinoma cell (OSCC) lines, radio-resistant cells, and patient-derived OSCC cells that served as a preclinical model.

## Results

### Inhibitory effect of BYL719 and BKM120 on OSCC and radioresistant OSCC cell lines

SCC4, SCC25, OML1, and radioresistant OML1-R cells were treated with BYL719 and BKM120 at indicated concentrations to evaluate the inhibitory effect and proper inhibitory concentrations of compounds. The results of cell viability showed that BKM120 significantly dose-dependently reduced the growth of SSC4 and SSC25 cells, and inhibited the growth of OML1 and OML1-R cells at 3 μM. BYL719 significantly dose-dependently inhibited the growth of SSC4 and SSC25 cells but not that of OML1 and OML1-R cells (Fig 1). The results indicated that pan-PI3K inhibitor BKM120 may be a better inhibitor against OSCC cell growth than BYL719.

### Enhanced radiosensitization effect of BYL719 and BKM120 on OSCC and radioresistant OSCC cells

OSCC (SCC4 and SCC25), OML1, and OML-R were pretreated with the inhibitors and then irradiated at 0 and 4 Gy to investigate whether BYL719 and BKM120 can sensitize OSCC cells to ionizing radiation (IR) and reduce IR-induced radioresistance. The combination treatment of BYL719 or BKM120 with 4-Gy IR significantly reduced the growth and colony formation of SCC25 cells compared with the single treatment of 4-Gy IR, BYL719, or BKM120 (Fig 2A). However, only BYL719 combined with 4-Gy treatment significantly reduced the growth of SCC4 cells (Fig 2B). In particular, BYL719 and BKM120 combined with 4-Gy treatment significantly inhibited the growth of OML1 and radioresistant OML1-R cells compared with single 4-Gy IR treatment (Fig 2C and 2D). To determine the effect of different radiation dose treated alone or combination treatments with BYL719 or BKM120 in cells growth, the SCC25 were treated with 2, 4, and 6-Gy IR treated alone or combination treatments with PI3K inhibitors. Results indicated that the treatment does-dependent reduced the growth of HNSCC cells (Fig 3).

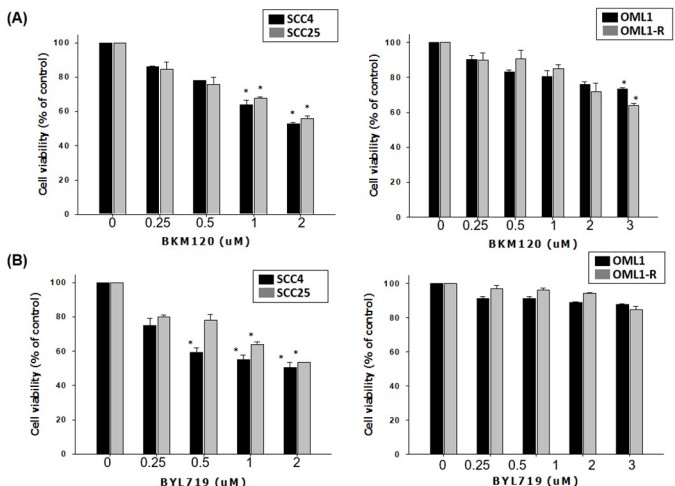

**Fig 1. Reduction effect of BKM120 and BYL719 on the growth of OSCC and radioresistant OML1-R cells.** SCC4, SCC25, and OML1-R cells were treated with BKM120 or BYL719 at indicated concentrations for 7 days. Cell viability was analyzed with MTS assay. Data are presented as mean ± SD, and the asterisks indicate significant difference. *P < 0.01 compared with the control group.

Whether or not the PI3K signaling inhibitor can increase the sensitivity of OSCC cells to IR treatment was confirmed as follows. Primary tumor cells were isolated from OSCC patients and then pretreated with BYL719 and BKM120 followed by 4-Gy IR treatment. Consistent with the above results, the combination treatment of PI3K inhibitors with IR showed approximately 18% reduction effect on the OSCC cells compared with the single treatment of IR (Fig 4). Therefore, the combination of PI3K inhibitors and IR exposure significantly enhanced the radiosensitivity of OSCC cells by reducing colony formation.

## Combination treatment effect of PI3K/mTOR pathway inhibitors with IR on the growth of OSCC cells

The PI3K/AKT/mTOR signaling pathway regulates cell proliferation. Increased radiosensitization can be observed in cells with inhibited PI3k/AKT/mTOR signaling pathway leading to cell cycle arrest [15]. OML-R cells were pretreated with PI3K/mTOR inhibitors combined with or without IR to evaluate the activity of mTOR inhibitor AZD2014 in the combination treatment with PI3K inhibitor or/and IR. As shown in Fig 5, AZD2014 combined with BKM120 or BYL719 treatment and 4-Gy IR showed an enhanced inhibitory effect on the growth of OML-R cells compared with each PI3K/mTOR inhibitor combined with IR. The inhibition percentages of triple combinations were 67% and 60% compared with the IR treatment. However, the combination treatment of AZD2014 with 4-Gy IR showed no increasing reduction effect compared with IR exposure only on OML-R cell growth.

## Discussion

Radiotherapy and surgery are both common treatment for patients with head and neck cancer, and the outcomes for either may is associated with the facility resources and provider expertise. In surgeries, the cancer tissue in local region and surrounding affected lymph node were removed, and the surgery treatment is commonly combined to adjuvant or neoadjuvant radiotherapy, chemotherapy or combined radio chemotherapy [16]. Patients with HNSCC treated with radiation therapy or chemotherapy is accompanied by severe side effects, including

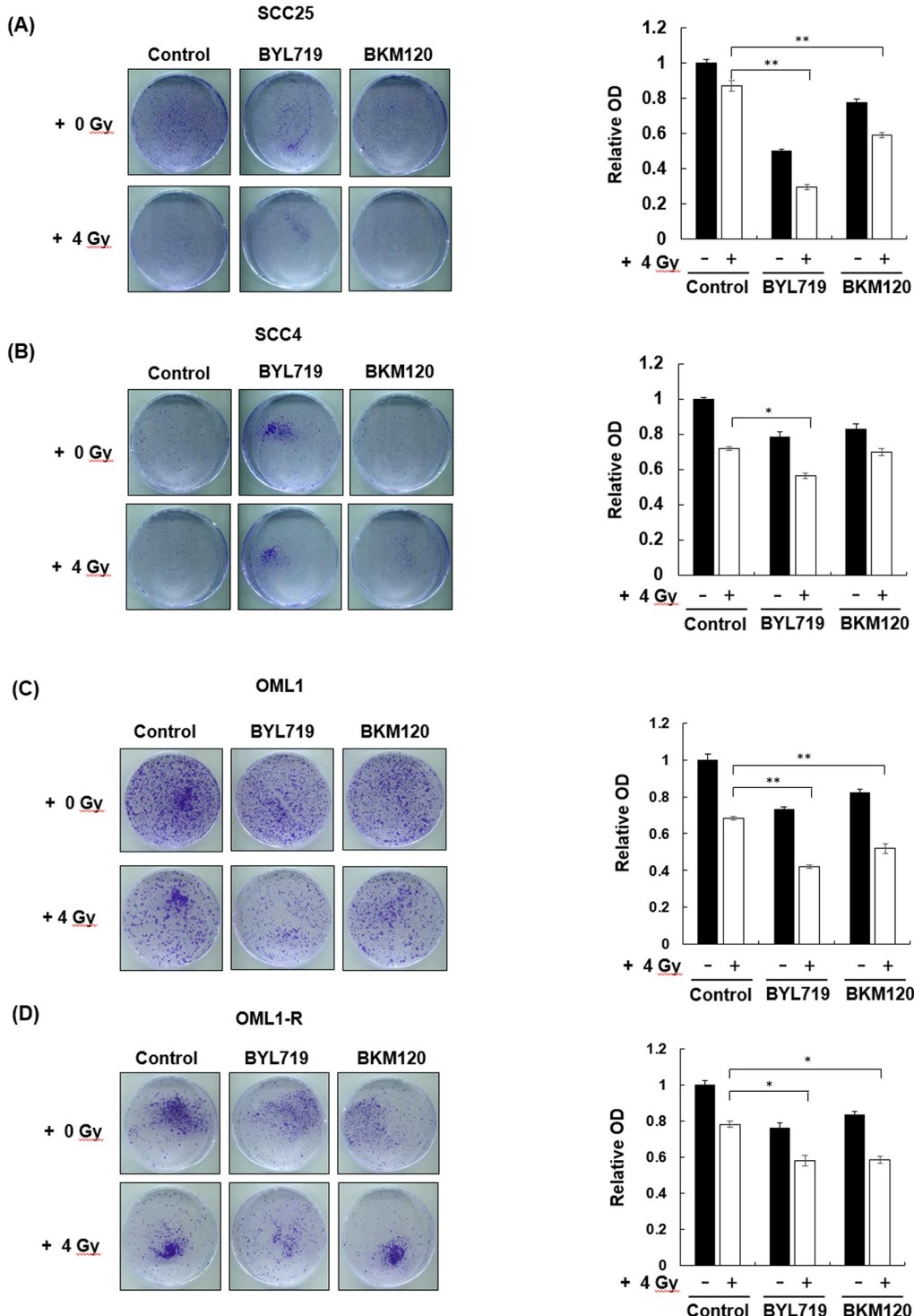

**Fig 2. Enhanced radiosensitivity upon IR exposure combined with BKM120 or BYL719 treatment in OSCC and radioresistant OML1-R cells.** SCC4, SCC25, and OML1-R cells were pretreated with BKM120 or BYL719 at 2 μM and then irradiated at 0 or 4 Gy. After 14 days, colonies were stained with crystal violet and quantified by an ELISA reader. Data are presented as mean ± SD, and the asterisks indicate significant difference. *P < 0.01 compared with the control group.

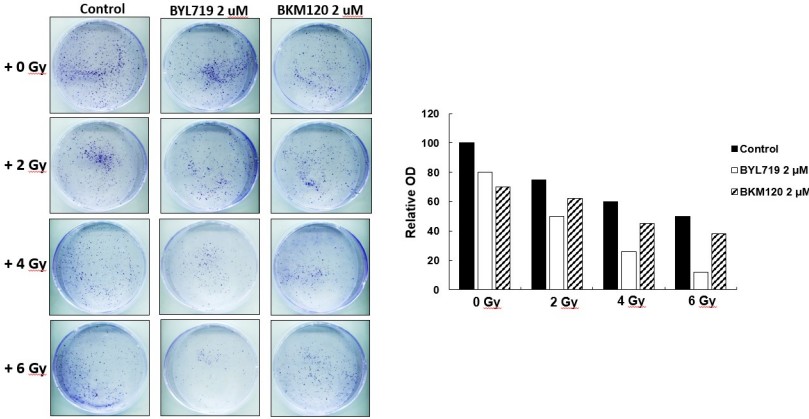

**Fig 3. IR exposure combined with BKM120 or BYL719 treatment dose-dependent reduced cell growth in OSCC cells.** SCC25 cells were pretreated with BKM120 or BYL719 at 2 μM and then irradiated at 0, 2, 4 or 6 Gy. After 14 days, colonies were stained with crystal violet and quantified by an ELISA reader. Data are presented as mean ± SD, and the asterisks indicate significant difference. *P < 0.01 compared with the control group.

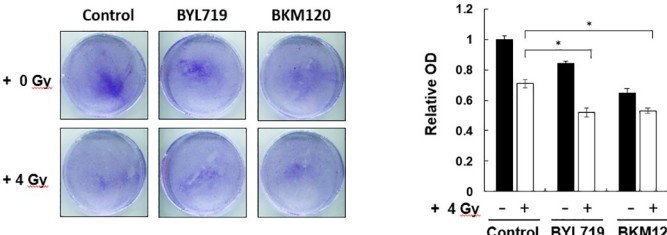

**Fig 4. Radiosensitivity enhancement effect of combination treatment of PI3K inhibitors and IR in patient-derived cells.** Patient-derived cells were derived from patients with OSCC. The cells were pretreated with BKM120 or BYL719 at 2 μM and then irradiated at 0 or 4 Gy. After 14 days, colonies were stained with crystal violet and quantified by an ELISA reader. Typical images of colony growth for the different treatments are shown. *P < 0.01 compared with the control group.

mucositis, dysphagia, leukopenia, and thrombocytopenia, which lead to increasing the risk of infection and bleeding resulting in affecting health-related quality of life [17]. Previous study revealed that combination treatment improved the overall survival of patients with advanced HNSCC via induction chemotherapy with cisplatin and 5 FU before radiotherapy or radio-chemotherapy, which not only increase the antitumor activity but also attenuate the administration concentration of treatment [18]. Present study revealed the combination treatment via PI3K inhibitor treatment before radiotherapy increased the antitumor effect in OSCC and patients-derived cells, which may provide information for the development of inhibitors with radio sensitization activity.

Radioresistance decreases the efficiency of radiotherapy, leading to HNSCC treatment failure. Therefore, overcoming the resistance problem has become a serious issue. In recent years, radiation sensitization was suggested to solve the challenges of raidoresistance through enhancing radiation damage to tumor tissue. A previous study showed that PI3k/AKT/mTOR signaling is an important mechanism associated with increasing radiation sensitivity in various cancer types [19, 20]. Our previous studies demonstrated that the mTOR inhibitors RAD001 and AZD2014 as well as the dual PI3K/mTOR inhibitor BEZ235 significantly induce radiation sensitivity by regulating cell cycle arrest, which may serve as potential small-molecular drugs

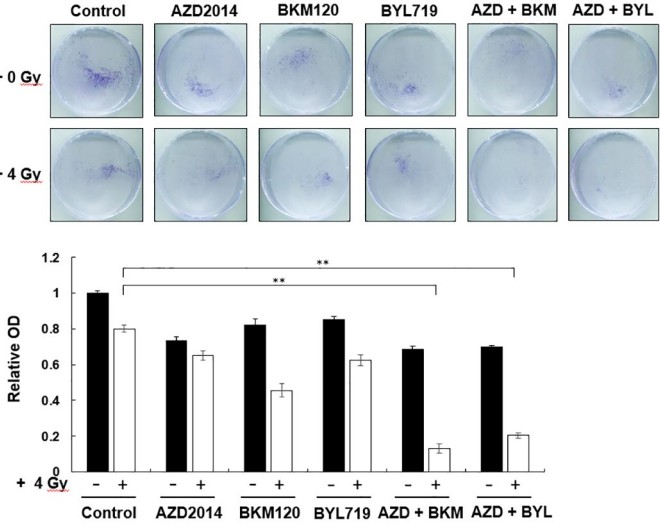

**Fig 5. Combination treatment with PI3K/mTOR inhibitors and IR increased radiosensitivity in OML1-R cells.**
The cells were treated with each PI3K/mTOR inhibitor alone or with the combination treatment of PI3K/mTOR inhibitors and IR. After 14 days, colonies were stained with crystal violet and quantified by an ELISA reader. Data are presented as mean ± SD, and the asterisks indicate significant difference. *P < 0.01 compared with the control group.

for patients with radioresistance [10]. In the present study, we used radioresistant oral cancer cells and patient-derived OSCC cells to assess the radiosensitivity of PI3K inhibitors. BKM120 and BYL179 enhanced the radiation sensitivity of OSCC cells.

Dysregulation of the PI3K/AKT/mTOR signaling pathway is a common mechanism to initiate the survival, proliferation, and metabolism of abnormal cells [21]. Therefore, drugs targeting PI3K or mTOR in clinical trials can effectively reduce tumor growth, and the combination treatment with radiation can improve anticancer activity [22]. Our previous study showed that the expression level of PI3K/AKT/mTOR-related proteins is higher in radioresistant OML1-R cells than in parental cell lines, suggesting that PI3K/AKT/mTOR signaling plays a critical role in radioresistance [10]. In the present study, the inhibitory effect of the combination treatment of PI3K inhibitor with radiation on radioresistant OML1-R cells was better than that of each single PI3K treatment.

Dual inhibition of mTORC1/mTORC2 activity enhances radiation-induced inhibition for the survival of OSCC cells. However, AZD2014 does not affect the survival of radiation-treated radioresistant OML1-R cells [23, 24]. Results showed that the triple combination treatment of AZD2014 and BKM120 or AZD2014 and BYL719 with radiation exhibited a significantly enhanced inhibitory effect on the growth of radioresistant OML1-R cells. The PI3K/mTOR signaling mechanism plays an important role in cellular mRNA translation and cell cycle progression. Therefore, further studies should investigate the mechanism underlying the inhibitory effect of the triple combination treatment on radioresistant cells. In conclusion, this study confirmed that PI3K/mTOR signaling is the important target for developing anticancer agents against radioresistant oral cancer cells.

## Materials and methods

### Ethics statement

All patients' personal information and medical records were collected, and the study was initiated after a formal approval of the institutional review board of the Dalin Tzu Chi Hospital,

Buddhist Tzu Chi Medical Foundation, Taiwan (approved number, B10302008). All experiments involved human samples obtained from patients, and informed consents were obtained from all subjects.

### Cell lines and reagents

SCC4 and SCC25, derived from a squamous cell carcinoma of the tongue, were purchased from the American Type Culture Collection (Manassas, VA, USA) and maintained in DMEM/F12 containing 10% fetal bovine serum (FBS), 1% penicillin–streptomycin, and 2 mM l-glutamine. OML1 and OML1-R (radioresistant) cells were established by Hon-Yi Lin and Michael W.Y. Chan et al. and cultured in RPMI1640 containing 10% FBS, 1% penicillin–streptomycin, and 2 mM l-glutamine. All cells were maintained in 37°C under a humidified atmosphere of 5% $CO_2$ with less than 4 months passage and checked for mycoplasma using kit (Lonza, Switzerland). BKM120 and BLY719 were provided by Novartis Pharmaceuticals Corporation (East Hanover, NJ, USA). AZD2014 was obtained from AstraZeneca (London, United Kingdom).

### Isolation and maintenance of primary cells

All patient-derived cells were isolated as previously described. In brief, the primary culture was attained from the tumor tissue of the upper lip of patients with OSCC. The cells were maintained in keratinocyte growth media (ScienCell Research Laboratories, Carlsbad, CA) with 15% FBS.

### Clonogenic assay

OSCC cells, radioresistant OML1-R cells, and patient-derived cells were pretreated with BKM120, BYL719, or AZD2014 and then irradiated at 0 or 4 Gy. After 14 days, the cells were stained and fixed with a mixed-well solution of 80% crystal violet and 20% methanol. Colonies (defined as group of >50 cells) were photographed with a digital camera. The cells were washed and added with 30% v/v acetic acid to completely dissolve the crystal violate, and the absorbance was detected at 595 nm by a Synergy 3 multimode microplate reader (BioTek, USA).

### Cell viability

The OSCC cell lines and radioresistant OML1-R cells were treated with BKM120 or BYL719 at indicated concentrations for 7 days, and the cell viability was determined through a colorimetric MTS assay in accordance with the manufacturer's instructions. The reagent-added 96 well plates were re-incubated at 37°C for 2 h, and the absorbance was detected at 480 nm by a Synergy 3 multimode microplate reader (BioTek, USA).

### Statistical analysis

Data are presented as mean ± standard deviation from at least three independent experiments. Statistically significant differences between the control group and treatment or IR-treated group and combination-treated group were analyzed using Student's t-test, and p-values less than 0.01 indicated significant differences. Data were analyzed using SPSS 20.0.

## Author Contributions

**Conceptualization:** Yu-Chieh Su.

**Data curation:** Fu-Cheng Chuang.

**Formal analysis:** Fu-Cheng Chuang.

**Investigation:** Chih-Chun Wang, Jian-Han Chen, Tzer-Zen Hwang, Shyh-An Yeh.

**Methodology:** Fu-Cheng Chuang, Chih-Chun Wang, Jian-Han Chen, Tzer-Zen Hwang.

**Supervision:** Yu-Chieh Su.

**Validation:** Tzer-Zen Hwang, Shyh-An Yeh.

**Writing – original draft:** Fu-Cheng Chuang, Yu-Chieh Su.

**Writing – review & editing:** Jian-Han Chen, Yu-Chieh Su.

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
