## [Decision Letter · Decision Letter 0]

14 Aug 2020

PONE-D-20-20798

PI3k inhibitors (BKM120 and BYL719) as radiosensitizers for head and neck squamous cell carcinoma during radiotherapy

PLOS ONE

Dear Dr. Su,

Thank you for submitting your manuscript to PLOS ONE. After careful consideration, we feel that it has merit but does not fully meet PLOS ONE’s publication criteria as it currently stands. Therefore, we invite you to submit a revised version of the manuscript that addresses the points raised during the review process.

Serious issues have been raised with respect to your manuscript.  Indeed, one reviewer suggested rejection.  Nevertheless, we would be willing to consider a revised manuscript if the authors feel they can address the criticisms.

We look forward to receiving your revised manuscript.

Kind regards,

Salvatore V Pizzo

Academic Editor

PLOS ONE

Journal Requirements:

2. Please provide additional information about each of the cell lines used in this work, including any quality control testing procedures (authentication, characterisation, and mycoplasma testing). For more information, please see http://journals.plos.org/plosone/s/submission-guidelines#loc-cell-lines

"NO - Include this sentence at the end of your statement: The funders had no role in

study design, data collection and analysis, decision to publish, or preparation of the

manuscript."

Reviewers' comments:

Reviewer's Responses to Questions

**Comments to the Author**

1. Is the manuscript technically sound, and do the data support the conclusions?

Reviewer #1: No

Reviewer #2: Partly

2. Has the statistical analysis been performed appropriately and rigorously? 

Reviewer #1: Yes

Reviewer #2: Yes

3. Have the authors made all data underlying the findings in their manuscript fully available?

Reviewer #1: Yes

Reviewer #2: Yes

4. Is the manuscript presented in an intelligible fashion and written in standard English?

Reviewer #1: No

Reviewer #2: Yes

5. Review Comments to the Author

Reviewer #1: The authors present an interesting analysis of the use of PI3K inhibitors to sensitize OSCC models to radiation therapy that has some novelty in this field. At this point, the manuscript lacks substantial supporting experiments to warrant publication, and some recommendations to help the authors improve the manuscript are listed below.

1) Inclusion of pharmacodynamic analysis of PI3K effector signaling in the models presented, with the doses used (which are often very high).

2) Inclusion of pharmacodynamic analysis of DNA damage responses (e.g. gH2AX, COMET assays) following radiation, as well as more detailed analysis of mechanisms of cell death.

3) In vivo modeling of the various combinations.

4) Supporting genetic approaches to model changes to PI3K isoforms and their regulation of the DNA damage response.

5) Evaluation of alternative modes of DNA damage to determine if the phenotype is unique to radiation and/or DNA double strand break induction.

Reviewer #2: This study describes the effects of the PI3K� inhibitors Alpelisib (BYL719) and Buparlisib (BKM120) on the radiation sensitivity of head and neck cancer cells. Although these studies are of some interest, they are limited in scope and descriptive. There are no mechanistic studies. In addition, the lack of in vivo studies makes it difficult to know if these agents and the combination with the mTOR inhibitor will translate into an actual treatment in patients.

1) Do the authors know if the concentrations used in the in vitro experiments are clinically relevant? What concentrations are found in patients receiving these drugs?

2) The use of a single radiation dose (Figures 2 and 3) give limited information on overall radiation sensitivity. This is usually best assessed by performing a full cell survival curve of control and 3-4 radiation doses.

3) Did the agents actually inhibit PI3K� under the conditions that were investigated for radiosensitization? This would be the best way to verify that the effects.

4) Why did the authors choose combine these inhibitors with an mTOR inhibitor.

5) The MTT like assay is useful for screening large number of conditions rapidly but is not as accurate as a clonogenic assay for detailed assessment of radiosensitization

6) The clinical applicability of these combinations is difficult to assess with in vitro experiments alone.

6. PLOS authors have the option to publish the peer review history of their article (what does this mean?). If published, this will include your full peer review and any attached files.

Reviewer #1: No

Reviewer #2: No

---

## [Author Response · Author response to Decision Letter 0]

2 Oct 2020

Review Comments to the Author

Reviewer #1: The authors present an interesting analysis of the use of PI3K inhibitors to sensitize OSCC models to radiation therapy that has some novelty in this field. At this point, the manuscript lacks substantial supporting experiments to warrant publication, and some recommendations to help the authors improve the manuscript are listed below.

1) Inclusion of pharmacodynamic analysis of PI3K effector signaling in the models presented, with the doses used (which are often very high).

Our response: As suggested by reviewer, we have considered the in vitro time-course pharmacodynamic analysis of the PI3K inhibitor, but we decided the assay will be performed in vivo study which may provide more information for the clinical trials. However, we still work on the development of the animal model because of the radiotherapy assay being not easy to perform.

2) Inclusion of pharmacodynamic analysis of DNA damage responses (e.g. gH2AX, COMET assays) following radiation, as well as more detailed analysis of mechanisms of cell death.

Our response: Previous study has demonstrated that BYL719 combined with RT induced the tumor γH2AX nuclear formation in HNSCC animal model (1). BKM120 also showed enhanced DNA damage activity in hepatocellular carcinoma cells (2). The detailed mechanisms we will perform in the HNSCC animal model in the further study.

Reference

1. Mizrachi A, Shamay Y, Shah J, Brook S, Soong J, Rajasekhar VK, et al. Tumour-specific PI3K inhibition via nanoparticle-targeted delivery in head and neck squamous cell carcinoma. Nature communications. 2017;8:14292.

2. Liu WL, Gao M, Tzen KY, Tsai CL, Hsu FM, Cheng AL, et al. Targeting Phosphatidylinositide3-Kinase/Akt pathway by BKM120 for radiosensitization in hepatocellular carcinoma. Oncotarget. 2014;5(11):3662-72.

3) In vivo modeling of the various combinations.

Our response: As suggested by reviewer, we understand the importance of animal model for the assessing the clinical applicability. The animal model commonly used the immunodeficiency animal model to assess the activity of the inhibitors, but the model is under the lack of host immunity-tumor cell interaction that may not truly reflect the activity of inhibitors in the in vivo assay. Furthermore, the application of radiotherapy in animal model is not easy to perform. Therefore, we are searching the best solution for the animal model in the further study.

4) Supporting genetic approaches to model changes to PI3K isoforms and their regulation of the DNA damage response.

Our response: As suggested by reviewer, the expression of γH2AX and PARP, which represent the DNA damage response, by western blotting will be checked in the combination treatment of PI3K inhibitor and radiotherapy in HNSCC animal model in the further study. 

5) Evaluation of alternative modes of DNA damage to determine if the phenotype is unique to radiation and/or DNA double strand break induction.

Our response: The alternative modes of DNA damage will be identified by performing NGS assay to comprehensively investigate the mechanism in the combination treatment of PI3K inhibitor and radiotherapy in HNSCC animal model in the further study. 

Reviewer #2: This study describes the effects of the PI3K� inhibitors Alpelisib (BYL719) and Buparlisib (BKM120) on the radiation sensitivity of head and neck cancer cells. Although these studies are of some interest, they are limited in scope and descriptive. There are no mechanistic studies. In addition, the lack of in vivo studies makes it difficult to know if these agents and the combination with the mTOR inhibitor will translate into an actual treatment in patients.

1) Do the authors know if the concentrations used in the in vitro experiments are clinically relevant? What concentrations are found in patients receiving these drugs?

Our response: Yes, we know the relevant between the in vitro experiments and clinical usage dose. The concentration of alpelisib used in patients is at a dose of 300 mg per day, and the concentration of buparlisib used in patients is 100 mg per day.

2) The use of a single radiation dose (Figures 2 and 3) give limited information on overall radiation sensitivity. This is usually best assessed by performing a full cell survival curve of control and 3-4 radiation doses.

Our response: The suggestion by reviewers was ideal to support more information. Our unshown results demonstrated that different radiation doses dose-dependent reduced the cell growth when treated alone or combination treatment with PI3K inhibitors. We added the result into the manuscript as figure 3. 

3) Did the agents actually inhibit PI3K� under the conditions that were investigated for radiosensitization? This would be the best way to verify that the effects.

Our response: The suggestion by reviewers was ideal to confirm the effect of PI3K inhibitor. The pan-PI3K inhibitor buparlisib (BKM120) and the PI3K inhibitor alpelisib (BYL719) are developed by Novartis Oncology for clinical patients, and the specific of the inhibitors against PI3K is definite under the conditions. 

4) Why did the authors choose combine these inhibitors with an mTOR inhibitor.

Our response: The PI3K/AKT/mTOR signaling pathway is a common mechanism to initiate the survival and proliferation. Therefore, the combination treatment of PI3K and mTOR inhibitor may boost the anticancer effect in the OSCC cells.

5) The MTT like assay is useful for screening large number of conditions rapidly but is not as accurate as a clonogenic assay for detailed assessment of radiosensitization

Our response: As suggested by reviewer, the MTT assay in present study was the primary experiments to determine the available concentrations of the inhibitors for the further assay. Using clonogenic assay to investigate the appropriate concentrations is more time consuming. 

6) The clinical applicability of these combinations is difficult to assess with in vitro experiments alone.

Our response: As suggested by reviewer, we understand the importance of animal model for the assessing the clinical applicability. The animal model commonly used the immunodeficiency animal model to assess the activity of the inhibitors, but the model is under the lack of host immunity-tumor cell interaction that may not truly reflect the activity of inhibitors in the in vivo assay. Furthermore, the application of radiotherapy in animal model is not easy to perform. Therefore, we are searching the best solution for the animal model in the further study.

---

## [Decision Letter · Decision Letter 1]

16 Nov 2020

PONE-D-20-20798R1

PI3k inhibitors (BKM120 and BYL719) as radiosensitizers for head and neck squamous cell carcinoma during radiotherapy

PLOS ONE

Dear Dr. Su,

Thank you for submitting your manuscript to PLOS ONE. After careful consideration, we feel that it has merit but does not fully meet PLOS ONE’s publication criteria as it currently stands. Therefore, we invite you to submit a revised version of the manuscript that addresses the points raised during the review process.

One reviewer is concerned that mechanistic studies are lacking and the study is purely descriptive.  Indeed this reviewer recommended rejection.  Nevertheless, we would consider a revised manuscript but these issues should be addressed. 

We look forward to receiving your revised manuscript.

Kind regards,

Salvatore V Pizzo

Academic Editor

PLOS ONE

Reviewers' comments:

Reviewer's Responses to Questions

**Comments to the Author**

1. If the authors have adequately addressed your comments raised in a previous round of review and you feel that this manuscript is now acceptable for publication, you may indicate that here to bypass the “Comments to the Author” section, enter your conflict of interest statement in the “Confidential to Editor” section, and submit your "Accept" recommendation.

Reviewer #2: (No Response)

Reviewer #3: All comments have been addressed

2. Is the manuscript technically sound, and do the data support the conclusions?

Reviewer #2: Partly

Reviewer #3: Yes

3. Has the statistical analysis been performed appropriately and rigorously? 

Reviewer #2: Yes

Reviewer #3: Yes

4. Have the authors made all data underlying the findings in their manuscript fully available?

Reviewer #2: Yes

Reviewer #3: Yes

5. Is the manuscript presented in an intelligible fashion and written in standard English?

Reviewer #2: Yes

Reviewer #3: Yes

6. Review Comments to the Author

Reviewer #2: The authors have responded to some comments but not some critical ones.

Comment 1) - the question was about concentrations and the authors have answered in mg

Comments 5 and 6). this reviewer agrees that performing clonogenic assays and animal experiments is time consuming. But they provide important information concerning the clinical applicability of this combination.

Reviewer #3: This study aimed to identify PI3K inhibitors that can enhance radiosensitivity in head and neck cancer cells. The authors showed that a combination of BKM120 or BYL719 with irradiation leads to an increase in radiosensitivity in HNSCC cells.

After a meticulous review of the manuscript, I found that the current study could be pertinent and informative for the readers this journal.

However, I think that there is some room for improvements before a further consideration for publication. At this point there is a need for minor revision before the manuscript could be reconsidered for publication in the journal.

According to my assessments, there are some points to clarify and revise:

In addition to radiotherapy, surgery is a widely used therapeutic option in patients with HNSCC. This should not remain unmentioned in the discussion, since surgery is superior to radiation therapy in early stages. Furthermore, in advanced tumors, radiotherapy is used in combination with chemotherapy and not as the sole modality. The authors should take a more differentiated view on the therapy of HNSCC and side effects of radiotherapy in the discussion section.

I highly recommend investigating the effects of BKM120 and BYL719 in vivo in the near future.

7. PLOS authors have the option to publish the peer review history of their article (what does this mean?). If published, this will include your full peer review and any attached files.

Reviewer #2: No

Reviewer #3: No

---

## [Author Response · Author response to Decision Letter 1]

23 Dec 2020

Review Comments to the Author

Reviewer #2: 

The authors have responded to some comments but not some

critical ones.

Comment 1) - the question was about concentrations and the authors

have answered in mg

Our response: As suggested by reviewer, the concentrations of the inhibitors in patients is associated with how many mg of the inhibitor that patient took, and the inhibitors concentration in the patient plasma appear dynamic which was showed as pharmacokinetics. For example, the alpelisib used in patients at a dose of 350 mg per day in cycle 1 showed the median maximum plasma concentration (Cmax) is 3490 ng/ml. The buparlisib used in patients at a dose of 100 mg per day in cycle 1 showed the Cmax is 1080 ng/ml. 

Comments 5 and 6). this reviewer agrees that performing clonogenic

assays and animal experiments is time consuming. But they provide

important information concerning the clinical applicability of this

combination.

Our response: As suggested by reviewer, we understand and agree that the animal model is important information for clinical applicability of the combination. However, previous studies had indicated that the PI3K/Akt/mTOR pathway inhibitors plus radiotherapy were ideal strategy for lung cancer and breast cancer (1, 2). Furthermore, several PI3K/Akt/mTOR inhibitors including alpelisib and buparlisib has been involved in clinical trials, indicating that the use of the inhibitor is available (3, 4). The combination treatment of PI3K/Akt/mTOR inhibitor and radiotherapy has also been identify that the strategy is clinical applicable(1). On the other hand, the use of the hospital facility was limited because of the novel coronavirus disease, which has influenced the progress of the study. In the near future, we will proceed the in vivo analysis to provide the direct evidence.

Reference:

1. Chen K, Shang Z, Dai AL, Dai PL. Novel PI3K/Akt/mTOR pathway inhibitors plus radiotherapy: Strategy for non-small cell lung cancer with mutant RAS gene. Life sciences. 2020;255:117816.

2. DuRoss AN, Neufeld MJ, Landry MR, Rosch JG, Eaton CT, Sahay G, et al. Micellar Formulation of Talazoparib and Buparlisib for Enhanced DNA Damage in Breast Cancer Chemoradiotherapy. ACS applied materials & interfaces. 2019;11(13):12342-56.

3. Rodon J, Brana I, Siu LL, De Jonge MJ, Homji N, Mills D, et al. Phase I dose-escalation and -expansion study of buparlisib (BKM120), an oral pan-Class I PI3K inhibitor, in patients with advanced solid tumors. Investigational new drugs. 2014;32(4):670-81.

4. Beck JT, Ismail A, Tolomeo C. Targeting the phosphatidylinositol 3-kinase (PI3K)/AKT/mammalian target of rapamycin (mTOR) pathway: an emerging treatment strategy for squamous cell lung carcinoma. Cancer treatment reviews. 2014;40(8):980-9.

Reviewer #3: 

This study aimed to identify PI3K inhibitors that can enhance radiosensitivity in head and neck cancer cells. The authors showed that a combination of BKM120 or BYL719 with irradiation leads to an increase in radiosensitivity in HNSCC cells. After a meticulous review of the manuscript, I found that the current study could be pertinent and informative for the readers this journal. However, I think that there is some room for improvements before a further consideration for publication. At this point there is a need for minor revision before the manuscript could be reconsidered for publication in the journal. According to my assessments, there are some points to clarify and revise: In addition to radiotherapy, surgery is a widely used therapeutic option in patients with HNSCC. This should not remain unmentioned in the discussion, since surgery is superior to radiation therapy in early stages. Furthermore, in advanced tumors, radiotherapy is used in combination with hemotherapy and not as the sole modality. Theauthors should take a more differentiated view on the therapy of HNSCC and side effects of radiotherapy in the discussion section. I highly recommend investigating the effects of BKM120 and BYL719 in vivo in the near future.

Our response: As suggested by reviewer, we added more discussion about the importance of surgery and the side effect when patients treated with radiotherapy or chemotherapy in the discussion section. We will proceed the in vivo analysis soon. Because of the novel coronavirus disease, the use of the hospital facility was limited that also influence the progress of the study.

---

## [Decision Letter · Decision Letter 2]

7 Jan 2021

PI3k inhibitors (BKM120 and BYL719) as radiosensitizers for head and neck squamous cell carcinoma during radiotherapy

PONE-D-20-20798R2

Dear Dr. Su,

We’re pleased to inform you that your manuscript has been judged scientifically suitable for publication and will be formally accepted for publication once it meets all outstanding technical requirements.

Kind regards,

Salvatore V Pizzo

Academic Editor

PLOS ONE

Additional Editor Comments (optional):

Reviewers' comments:

Reviewer's Responses to Questions

**Comments to the Author**

1. If the authors have adequately addressed your comments raised in a previous round of review and you feel that this manuscript is now acceptable for publication, you may indicate that here to bypass the “Comments to the Author” section, enter your conflict of interest statement in the “Confidential to Editor” section, and submit your "Accept" recommendation.

Reviewer #2: (No Response)

Reviewer #3: All comments have been addressed

2. Is the manuscript technically sound, and do the data support the conclusions?

Reviewer #2: Yes

Reviewer #3: Yes

3. Has the statistical analysis been performed appropriately and rigorously? 

Reviewer #2: Yes

Reviewer #3: Yes

4. Have the authors made all data underlying the findings in their manuscript fully available?

Reviewer #2: Yes

Reviewer #3: Yes

5. Is the manuscript presented in an intelligible fashion and written in standard English?

Reviewer #2: Yes

Reviewer #3: Yes

6. Review Comments to the Author

Reviewer #2: (No Response)

Reviewer #3: (No Response)

7. PLOS authors have the option to publish the peer review history of their article (what does this mean?). If published, this will include your full peer review and any attached files.

Reviewer #2: No

Reviewer #3: No

---

## [Editor Report · Acceptance letter]

11 Jan 2021

PONE-D-20-20798R2 

PI3k inhibitors (BKM120 and BYL719) as radiosensitizers for head and neck squamous cell carcinoma during radiotherapy 

Dear Dr. Su:

I'm pleased to inform you that your manuscript has been deemed suitable for publication in PLOS ONE. Congratulations! Your manuscript is now with our production department. 

Kind regards, 

on behalf of

Dr. Salvatore V Pizzo 

Academic Editor

PLOS ONE